# The Impact of the COVID-19 Pandemic on Depressive Disorder with Postpartum Onset: A Cross-Sectional Study

**DOI:** 10.3390/healthcare11212857

**Published:** 2023-10-30

**Authors:** Livia Ciolac, Marius Lucian Craina, Virgil Radu Enatescu, Anca Tudor, Elena Silvia Bernad, Razvan Nitu, Lavinia Hogea, Lioara Boscu, Brenda-Cristiana Bernad, Madalina Otilia Timircan, Valeria Ciolac, Cristian-Octavian Nediglea, Anca Laura Maghiari

**Affiliations:** 1Department of Obstetrics and Gynecology, Faculty of General Medicine, “Victor Babes” University of Medicine and Pharmacy Timisoara, 300041 Timisoara, Romania; livia.ciolac@umft.ro (L.C.); mariuscraina@hotmail.com (M.L.C.); bernad.elena@umft.ro (E.S.B.); nitu.dumitru@umft.ro (R.N.); timircan.madalina@yahoo.com (M.O.T.); 2Doctoral School, Faculty of General Medicine, “Victor Babes” University of Medicine and Pharmacy Timisoara, 300041 Timisoara, Romania; lioara.boscu@umft.ro; 3Clinic of Obstetrics and Gynecology, “Pius Brinzeu” County Clinical Emergency Hospital, 300723 Timisoara, Romania; 4Center for Laparoscopy, Laparoscopic Surgery and In Vitro Fertilization, “Victor Babes” University of Medicine and Pharmacy Timisoara, 300041 Timisoara, Romania; 5Psychiatric Clinic, “Pius Brinzeu” Emergency County Hospital, 156 Liviu Rebreanu Blvd., 300723 Timisoara, Romania; enatescu.virgil@umft.ro; 6Department of Neuroscience, “Victor Babes” University of Medicine and Pharmacy Timisoara, 2 Eftimie Murgu Square, 300041 Timisoara, Romania; hogea.lavinia@umft.ro; 7Discipline of Computer Science and Medical Biostatistics, University of Medicine and Pharmacy, 300041 Timisoara, Romania; atudor@umft.ro; 8Research Center in Dental Medicine Using Conventional and Alternative Technologies, School of Dental Medicine, “Victor Babes” University of Medicine and Pharmacy of Timisoara, 9 Revolutiei 1989 Ave., 300070 Timisoara, Romania; 9Center for Neuropsychology and Behavioral Medicine, “Victor Babes” University of Medicine and Pharmacy, 300041 Timisoara, Romania; 10Senate Office, “Victor Babes” University of Medicine and Pharmacy, Eftimie Murgu Square, No. 2, 300041 Timisoara, Romania; 11Department of Sustainable Development and Environmental Engineering, University of Life Sciences “King Michael I”, 300645 Timisoara, Romania; valeriaciolac@usvt.ro; 12Clinic of Anaesthesia and Intensive Care, Emergency County Hospital “Pius Brinzeu”, 325100 Timisoara, Romania; nedigleacristian@gmail.com; 13Department of Anatomy and Embryology, “Victor Babes” University of Medicine and Pharmacy Timisoara, 300041 Timisoara, Romania; boscu.anca@umft.ro

**Keywords:** postpartum depression, screening, pandemic, COVID-19, Edinburgh Postnatal Depression Rating Scale (EPDS)

## Abstract

Background: COVID-19 has led to a global health crisis that is defining for our times and one of the greatest challenges to emerge since World War II. The potential impact of the pandemic on mental health should not be overlooked, especially among vulnerable populations such as women who gave birth during the COVID-19 pandemic. Materials and Methods: The study is a cross-sectional survey conducted from 1 March 2020 to 1 March 2023, during the period of the SARS-CoV-2 (COVID-19) pandemic, based on a retrospective evaluation of 860 postpartum women. The screening tool used to assess symptoms of postpartum depression was the Edinburgh Postnatal Depression Rating Scale (EPDS) questionnaire. The questionnaire was completed both in the Obstetrics and Gynaecology Clinical Sections I and II of the “Pius Brînzeu” County Emergency Hospital in Timisoara, Romania, and online using Google Forms. Results: The highest severity of postpartum depression symptoms was observed during the COVID-19 pandemic. The results of the study conducted during the period of the SARS-CoV-2 pandemic (COVID-19) showed that the prevalence of major postpartum depressive disorder (EPDS ≥ 13) was 54.2% (466 patients), while 15.6% (134) had minor depressive disorder (10 < EPDS ≤ 12) in the first year after delivery. Comparing these results with those obtained in research conducted before the onset of the pandemic period showed an alarming increase in the prevalence of postpartum depression. The risk factors associated with postpartum depression included the type of delivery, level of education, socio-economic conditions, health status, age, background, and personal obstetric history (number of abortions on demand, parity). Conclusions: The effects of the pandemic on mental health are of particular concern for women in the first year after childbirth. Observing these challenges and developing effective measures to prepare our health system early can be of great help for similar situations in the future. This will help and facilitate effective mental health screening for postpartum women, promoting maternal and child health.

## 1. Introduction

The 2019 coronavirus disease (COVID-19), caused by severe acute respiratory syndrome coronavirus 2 (SARS-CoV-2), was first recognized in December 2019 in Wuhan, the capital of China’s Hubei province. Since then, the disease has spread worldwide, leading to a coronavirus pandemic [1]. In January 2020, it was recognized by the World Health Organization (WHO) as a major public health concern [1]. COVID-19 has led to a global health crisis that is defining for our times and one of the greatest challenges to emerge since World War II. As a result of this sudden onset epidemic, governments and public health authorities urgently needed guidance and useful information on effective interventions to protect the health of the population [2].

Given the paucity of data in the existing academic literature on clinically manifested peripartum depression among women who gave birth during the COVID-19 pandemic, we set out to assess the risk of developing peripartum depressive disorder during the COVID-19 pandemic compared with the risk among women who gave birth before the onset of the COVID-19 pandemic.

The postpartum period represents a time of increased vulnerability for the development of psychiatric disorders [2]. The potential impact of the pandemic on mental health should not be overlooked, especially among vulnerable populations [3,4]. According to the World Health Organization, approximately 10% of parturients and 13% of postpartum women experience some form of mental disorder, mainly depression [5]. In the context of the COVID-19 pandemic, maternal distress may be exacerbated by concerns and fears about the risk of infection or hospitalization due to the coronavirus, given that perinatal morbidity and mortality associated with COVID-19 have been documented [6,7].

Postpartum depression is defined in psychiatric nomenclature as a major depressive disorder with a specific onset in the first month after parturition, with the possibility of extending the time interval up to one year [8]. It is considered a major public health problem as it affects both the mother and the child and has a high prevalence globally, ranging from 10% to 20% in most studies [9]. The term postnatal depression is generically used in the literature to designate the picture of depressive symptoms, with an onset in the period following childbirth and whose etiology is related to childbirth, as well as to hormonal (physiological), psychological, environmental, or social aspects, which occur in temporal proximity to the moment of birth [9].

Women who develop postnatal depression are at greater risk of relapsing during subsequent pregnancies and of developing major depressive disorder outside the perinatal period [8]. Studies in recent years have shown that the nature of the early mother–infant relationship in the context of postpartum depression is predictive of the child’s cognitive, emotional, and social development [10].

Previous publications have noted an increased likelihood of depressive symptoms among pregnant or postpartum women, but these studies have been limited by the relatively small sample sizes of the patients included, as well as the individual particularities of each country in which they were conducted [11,12]. The extent to which parturients were emotionally affected by the pandemic remains unclear and problematic. It is therefore necessary to clarify which women are more at risk of being affected by peripartum depression. It is also important that the factors involved in the onset of mental distress are identified to help develop effective screening and prevention strategies among vulnerable populations.

The aim of this study was to assess the depressive symptomatology of postpartum women in pandemic times and to investigate potential associations between the symptoms of depressive disorder onset in close temporal proximity to the time of birth and the socio-demographic conditions, health status, and obstetric particularities of the patients.

## 2. Materials and Methods

### 2.1. Sample Description

The study is a cross-sectional survey conducted from 1 March 2020 to 1 March 2023, during the SARS-CoV-2 (COVID-19) pandemic, based on a retrospective evaluation of 860 postpartum women. This manuscript of the conducted observational study was prepared following STROBE guidelines [13]. We performed a G*Power (version 3.1.9.7) test for the chi-square test family, contingency tables as goodness of fit tests with 90% power, 0.05 level of significance, one degree of freedom, and 0.11 as an effect size. The estimated sample size was 853 respondents.

For the purpose of our study, the patients were meticulously screened and chosen based on a comprehensive set of inclusion and exclusion criteria, to ensure the specificity and uniformity of our participant pool. All 860 postpartum women were eligible participants incorporated in the investigation. The study was carried out in the Obstetrics and Gynaecology Clinical Sections I and II of the “Pius Brînzeu” County Emergency Hospital in Timisoara, Romania.

#### Inclusion and Exclusion Criteria

Participants were enrolled into the study if they met the following inclusion criteria:Delivered mothers within age group of 18–50 years;Women who had given birth in the last year from the date of completion of the survey;No history of psychiatric disorders;No history of peripartum depression in previous pregnancies;No past incidents or diagnoses of COVID-19 infection in the last year;Women who expressed an interest in this topic and who have provided informed consent to participate in the study.

Participants were excluded if they met any of the following conditions:Women with current and past use of psychotropic medications;Women with a high-risk pregnancy (including preeclampsia, gestational diabetes mellitus, chronic disease, intrauterine growth restriction, known fetal anomalies, or chromosomal aberrations);Women who had a history of psychiatric disorders or mental health issues.

Prior informed consent was obtained for each patient since it consisted of sensitive data and a vulnerable group of study participants.

The screening tool used to assess the symptoms of postpartum depression was the Edinburgh Postnatal Depression Rating Scale (EPDS) questionnaire. The questionnaire was completed both in the Obstetrics and Gynaecology Clinical Departments I and II and online using Google Forms. The percentages of participants that completed the EPDS questionnaire in the hospital was 93.26% (802), while 6.74% (58) completed it online.

In order to highlight the particularities of postpartum depression, as well as the factors favoring this pathology, in the patients included in this study, the following parameters were also taken into account: age; marital status; background; level of education; working conditions (risk at work); socio-economic conditions; health status; personal pathological history; parity; the method of obtaining pregnancy; type of birth, under the recommendation of the medical consultant; the mother’s wishes regarding the type of birth; the number of miscarriages; and the number of abortions upon request.

### 2.2. Ethics Declarations

The Local Commission of Ethics for Scientific Research from the Timis County Emergency Clinical Hospital “Pius Brînzeu” in Timisoara, Romania, operates under the article 167 provisions of Law no. 95/2006, art. 28, chapter VIII of order 904/2006; with EU GCP Directives 2005/28/EC, International Conference of Harmonisation of Technical Requirements for Registration of Pharmaceuticals for Human Use (ICH); and with the Declaration of Helsinki—Recommendations Guiding Medical Doctors in Biomedical Research Involving Human Subjects. The current study was conducted according to the guidelines of the Declaration of Helsinki, followed the European Union General Data Protection Regulation (GDPR), and was approved by the Local Commission of Ethics for Scientific Research from the Timis County Emergency Clinical Hospital “Pius Brinzeu” in Timisoara, Romania, No. 184/10.02.2020.

### 2.3. Edinburgh Postnatal Depression Scale Questionnaire

Depression is a pathology where psychometric assessment is particularly useful in confirming the diagnosis.

The Edinburgh Postnatal Depression Rating Scale (EPDS) questionnaire is one of the most widely used screening tools for assessing symptoms of perinatal depression and anxiety [14]. The EPDS had been validated against clinical diagnoses in over 37 languages since its development, and it has been regarded as the most frequently used and well-validated screening tool for postpartum depression [15]. The EPDS is a 10-item questionnaire, which has been validated in different countries and populations, including Romania [16,17,18]. It is a simple instrument that assesses emotional experiences over the past seven days using 10 Likert scale questions, is easy to fill in and interpret, requires no specialist psychiatric expertise, and could easily be incorporated into the health care services offered to all women in the postnatal period [14].

The Edinburgh Postnatal Depression Scale (EPDS) has become a leading choice for identifying women at risk for the diagnosis [19]. This self-report tool was developed and tested in health centers in Edinburgh and Livingston (UK) by Cox, Holden, and Sagovsky in 1987 to help detect women suffering from postnatal depression [20]. Since its inception, the EPDS has been adapted for use in several countries and has become the most widely used tool for assessing postpartum depression [19].

The ratings of responses to the 10 questions are summed, and the resulting score can assess the likelihood that the patient has clinical depression. The EPDS scale is composed of three structural factors: the “depression” factor through questions 1, 2, and 8; the “anxiety” factor through questions 3, 4, and 5; and the “suicide” factor through question 10. A score >10 betrays a possible depression (minor depressive disorder), and a score ≥13 suggests a major depressive disorder (moderate to severe) [14,21,22].

The EPDS scale can provide stable results, especially when assessments are performed repeatedly. In comparison with a clinical diagnostic interview, the EPDS demonstrated the following psychometric properties: a specificity of 78%, a sensitivity of 86%, and a positive predictive value of 73% for women scoring >10 [20]. Validity studies show that the scale can correctly identify 92.3% of women with postpartum depression [20].

### 2.4. Statistical Assessment

The collected data were introduced in Excel format and statistically processed with the SPSSv.17 software package.

Nominal variables were represented as frequency tables for which percentage distributions (pie) were plotted and associated with the χ2 (Chi square) test of concordance. For the numeric variables, indicators of central tendency (mean and median) and dispersion (standard deviation and standard error of the mean) were calculated, and for the study of the association between them, a Spearman’s nonparametric linear correlation analysis was carried out, with the help of which we calculated the correlation coefficients and probability values that provided us the significance of the correlation (*p*-values must be below 0. 05 for the association to be significant). For comparisons between two sets of numerical variables, the Mann–Whitney U nonparametric test was used, and for comparisons between more than two sets, the Kruskal–Wallis nonparametric test was applied.

## 3. Results

The study involved 860 women in their first year after childbirth. The distribution of socio-demographic characteristics and obstetric indicators among study participants are presented in Table 1.

Following the assessment of the clinical status of the women, using the Edinburgh Psychiatric Postnatal Depression Rating Scale (EPDS), 54.2% (466 patients) had major depressive disorder, 15.6% (134) had minor depressive disorder, and 30.2% (260 patients) had no depressive disorder.

The recorded values of the Edinburgh score following the completion of the questionnaire ranged from 0 to 28, with a mean score of 13.06. The maximum possible Edinburgh score of 30 was not recorded in this study (Figure 1).

Ratings of the responses to the 10 questions of the EPDS questionnaire are evidenced in Table 2.

We aimed to analyze the prevalence of suicidal ideation in postpartum women in the context of the ongoing pandemic of the coronavirus disease. We found that 14.9% (247) of 860 mothers who completed the EPDS question related to suicidal ideation (question 10 of the questionnaire) reported some suicidal ideation: 2.7% (23) reported that the thought of harming themselves had occurred to them quite often, and 12.2% (105) reported that it sometimes occurred to them.

Of the 860 patients, 57.7% (496) gave birth by caesarean section, and 42.3% (364) gave birth naturally. As for the mother’s wishes regarding the type of birth, 71.9% (618) would have liked to give birth naturally, and 28.1% (242) would have opted for caesarean section. A significant association was found between the mother’s desire for future birth and the type of birth (Chi square, *p* < 0.001); in other words, the obstetrician took the mother’s desire into account when determining the type of birth. The proportion of births by caesarean section, for mothers who wanted caesarean section, was significantly increased compared with the proportion of caesarean sections for mothers who wanted natural birth.

A significant association was established between type of delivery and depressive disorder (Chi square, *p* = 0.003). The proportion of mothers without depressive disorder was significantly increased among those who delivered naturally (Chi square, *p* = 0.0012), and the proportion of those with major depressive disorder was significantly decreased for mothers who delivered naturally (Chi square *p* = 0.0041) (Table 3). The occurrence of postpartum depressive disorder was significantly influenced by the type of delivery.

There was a significant association between the type of birth and marital status (Chi square, *p* < 0.001). Married mothers who gave birth by Caesarean section were significantly more numerous than those who gave birth naturally (Chi square, *p* = 0.038). Those living in cohabitation who gave birth by Caesarean section were significantly fewer than those who gave birth naturally (Chi square, *p* = 0.0011). Single mothers who gave birth by caesarean section were significantly more than those who gave birth naturally (Chi square, *p* = 0.026). (Table 3)

The association between the type of birth and the level of education was significant (Chi square, *p* < 0.001). The proportion of mothers without a high school education was significantly increased among those who gave birth naturally (Chi square, *p* < 0.001), while the proportion of mothers with higher education was significantly increased among those who gave birth by caesarean section (Chi square, *p* = 0.012). (Table 3)

The proportion of mothers who have a high school education was significantly increased among those with postpartum depression (*p* = 0.010), while the proportion of those with higher education was significantly increased among those without postpartum depressive disorder (*p* = 0.028) (Table 3). The association between the education level and postpartum depression was significant; higher education seemed to be a protective factor against the onset of depressive symptomatology (Chi square, *p* = 0.028).

The association between depressive disorder and health status was significant (Chi square, *p* < 0.001). The proportion of mothers with depressive disorder was significantly increased among those with fair or poor health status (Chi square, *p* < 0.001) (Table 3).

A direct, significant, and weak correlation was found between the number of miscarriages and the number of births (Spearman correlation coefficient r = 0.165, *p* < 0.001)—women who had an increased number of miscarriages also had an increased number of births.

A direct, significant and weak correlation was found between the number of abortions on demand and the number of births (Spearman correlation coefficient r = 0.142, *p* < 0.001)—women who had an increased number of abortions on demand also had an increased number of births.

A direct, significant, and weak correlation (Spearman correlation coefficient r = 0.138, *p* = 0.001) was found between the number of abortions on demand and the Edinburgh score—women who had an increased number of abortions on demand also had an increased Edinburgh score.

The maternal age was significantly lower for the respondents with depressive disorder (Mann–Whitney U nonparametric test, *p* = 0.025), indicating a possible association of younger age with the onset of postnatal depression (Table 4 and Figure 2).

Women with postnatal depressive disorder had a significantly increased number of abortions on demand (Mann–Whitney U nonparametric test, *p* = 0.005).

The number of abortions on demand was significantly increased among mothers with depressive disorder, both minor and major; therefore, the personal obstetric history of patients may be a factor in the development of postnatal depressive disorder (Kruskall–Wallis nonparametric test, *p* = 0.018) (Table 4 and Figure 3).

Mothers who gave birth by caesarean section had a significantly increased Edinburgh score (Mann–Whitney U nonparametric test, *p* = 0.008), which meant that they were more likely to develop depressive symptoms (Table 5).

## 4. Discussion

The results of the study conducted during the period of the SARS-CoV-2 pandemic (COVID-19) showed that the prevalence of major postpartum depressive disorder was 54.2% (466 patients), while 15.6% (134) had minor depressive disorder in the first year after delivery. Comparing these results with those obtained in research conducted before the onset of the pandemic period showed an alarming increase in the prevalence of postpartum depression. The incidence of postpartum depressive disorder worldwide in the non-pandemic period was about 10% in developed countries and about 21–26% in developing countries [23,24]. Previous research has also found that during natural disasters that struck humanity, the prevalence rates of mental disorders among postpartum women were significantly higher than those among the general population [25]. Moreover, studies conducted in the pre-pandemic period have shown that in about 30% of patients with postnatal depressive disorder, the recovery or resolution of depressive symptoms may take more than 1 year, but it is not known to what extent the pandemic may influence the recovery time, and further research is needed.

It is known that postnatal depression has long-term consequences for both the mother and the infant, so identifying the risk factors involved could help to conduct targeted screening, as well as to design targeted intervention strategies to prevent the long-term impact of the pandemic on maternal mental health and infant development [26,27].

In terms of the mothers’ backgrounds, 69.5% (598) were from urban areas, and 30.5% (262) were from rural areas. The implemented preventive measures, quarantine, home isolation, social distancing, aimed at stopping further spread of the virus, have increased the level of anxiety and stress among postpartum women [28,29], especially for mothers living in urban areas [30], regarded as high-incidence areas [31].

The occurrence of postpartum depressive disorder was significantly influenced by the type of birth. Of the 860 patients included in the study, 57.7% (496) gave birth by caesarean section, and 42.3% (364) gave birth naturally. In terms of the mother’s wishes regarding the type of birth, 71.9% (618) would have preferred a natural birth, and 28.1% (242) would have opted for a caesarean birth. The obstetrician also considered the mother’s wishes when deciding on the type of birth. The proportion of births by caesarean section, for mothers who wanted caesarean sections, was significantly increased compared with the proportion of caesarean sections, for mothers who wanted to give birth naturally during the period of our study.

Mothers who gave birth during the pandemic reported a higher level of perceived pain during labor [32] or during the recovery period secondary to the caesarean section. A significant association between the type of birth and depressive disorder was established in our own study; the proportion of mothers without depressive disorder was significantly increased among those who gave birth naturally, while the proportion of mothers with major depressive disorder was significantly decreased for mothers who gave birth naturally. In addition, patients who gave birth by caesarean section or those who experienced perinatal complications [33] tended to be more depressed, as the length of hospital stay increased [34]. It is important to mention that the presence of cardiovascular risk factors, with a negative impact on pregnancy, can also affect the mother’s physical well-being; therefore, it is crucial to identify and properly treat these factors for optimal maternal health [35]. These aspects seem to be responsible for the increased risk of postpartum depression.

In contrast to the findings in most previous studies, single mothers did not show symptoms of postpartum depression to a significantly greater extent than mothers who were married or cohabiting [36]. By marital status, 86.7% (746) of the mothers included in the study were married, 11.4% (98) were cohabiting, and 1.9% (16) were single. Single-mother families often face structural disadvantages due to having lower income and less time together with their children [36]. In a Swedish study, children of single-parent households (90% women) were found to be at increased risk for childhood psychopathology, suicide attempts, and drug addiction [37]. Single mothers may face not only the non-shared care of a child but also economic problems resulting from discriminatory wage levels and the absence of a second income from a partner [38]. Paternal support is known to play an important role in the postnatal period but was not interpreted during this study.

In accordance with previous studies, the results indicated that giving birth at a young age was associated with symptoms of postpartum depression [36]. Teenage mothers were at increased risk for depression [39]. Several factors might be of importance for this finding. First, an adolescent mother faces the challenge of her own developmental tasks, in addition to the challenge of taking care of a newborn [36]. Second, early motherhood is associated with lower degrees of education and lower income [40]. Childbirth during adolescence is demanding as it takes place during an intense mental and physical developmental stage, challenging or forcing the transition from childhood to adulthood [36]. Children of teenage mothers have been shown to have delays in cognitive and language abilities [41].

The COVID-19 pandemic has increased women’s insecurity from many perspectives. The present study also investigated the risk of postnatal depression according to the mothers’ background, education level, and socio-economic conditions, and these variables were found to have a greater impact on depressive symptoms compared with younger age or unmarried status. Women with a good standard of living appeared to have a lower risk of experiencing postnatal depression. Research conducted between 2020 and 2021 highlighted an increased prevalence of food insecurity due to negative changes in food availability [42]. A lack of food security negatively influences quality of life and, thus, health status and is associated with poor nutrition for pregnant or breastfeeding women, obesity, depression, and even high mortality rates [43]. Post-partum depression, when associated with food insecurity found among families with poor living conditions, increases the risk of delayed early child development [44]. Families with poor living conditions should therefore be identified and supported to prevent maternal mental health problems.

The study showed that postpartum depression was influenced by parity. Of the patients, 65.2% were primiparous. Multiparous mothers were less likely to experience postpartum depression, compared with primiparous mothers. According to a study in the peer-reviewed literature, 50–60% of women experience postpartum depression after their first birth [45]. Also, a study conducted in Japan identified a positive correlation between perinatal depressive disorder and primiparity [46]. The experience of the first child may cause more fear and anxiety about childbirth or the immediate postpartum period; therefore, the support of these women could contribute to the prevention and timely diagnosis of depressive disorder with onset close to parturition.

The relationship between suicidal thoughts and suicidal acts in the postpartum period is not clear, but it is prudent to assume that suicidal thoughts are a marker of an increased risk of suicide [47]. We aimed to analyze the prevalence of suicidal ideation in postpartum women, in the context of the ongoing pandemic of coronavirus disease. Suicidal ideation was defined as an answer of “sometimes” or “yes, quite often” to question 10 of the EPDS, “The thought of harming myself has occurred to me”; no suicidal ideation was defined by answering “hardly ever” or “never” for question 10 [48]. We found that 14.9% (247) of 860 mothers who completed the EPDS question related to suicidal ideation (question 10 of the questionnaire) reported some suicidal ideation: 2.7% (23) reported that the thought of harming themselves had occurred to them quite often, and 12.2% (105) reported that it occurred to them sometimes. Like all previous studies, the main limitation of the EPDS suicidality measure is that it is about self-reported thoughts of self-harm. Self-reporting of suicidal thoughts may lead to under-reporting of them; also, it is possible that self-reporting of these ideations may more accurately reflect the truth problem than clinical interviews. However, healthcare professionals, using the EPDS, should be aware of the significant suicidality that is likely to be present in women endorsing “yes, quite often” to question 10 of the EPDS.

Although the original purpose of the EPDS was postpartum depression screening, recent studies revealed that subscales of the EPDS can be used in new ways, such as anxiety disorder screening [15,49]. Given that postpartum depression is often accompanied by anxiety [50], this means that the application of its subscales can help evaluate mothers’ mental health conditions in greater detail [15]. Thus, the “anxiety” factor of the EPDS scale, known as a subscale of EPDS, consisting of items 3, 4, and 5 (EPDS-3A) of the questionnaire, may help to identify anxiety symptoms, expected to be accentuated in the context of the COVID-19 pandemic and the fact that the birth took place in the hospital, where patients that tested positive for COVID-19 infection were being treated, with the corresponding increased risk of infection for the rest of the hospitalized mothers. Table 2 shows the percentage distribution of the responses to questions 3, 4, and 5 of the EPDS questionnaire. It is noteworthy that the percentage distribution recorded for score 3 of questions 3 and 4 of the EPDS questionnaire was higher compared with the corresponding one for the rest of the questions. In addition to hospitalization, the fear of adverse effects of the virus and vaccines on the developing fetus; disruption of maternal–infant bonding; social isolation; less contact with friends, family, and social care services; and financial problems related to lockdown measures exerted additional anxiety in the pandemic context [51].

A cutoff of ≥5 on the EPDS-3A score was found to be efficient for identifying women experiencing clinical levels of anxiety (sensitivity: 70.9%; specificity: 92.2%) [52]. In settings where the EPDS is already implemented and where adding extra mental health screening instruments is not feasible, the EPDS-3A could be used as a resource-effective means of detecting mothers with possible anxiety disorder [52]. The vast majority of women screening positive on the EPDS-3A also screen positive on the total EPDS; using the EPDS-3A score along with the total EPDS score can indicate whether a mother may be suffering from anxiety either co-morbid with depression or as the primary problem [52].

The conducted research supports existing studies on the prevalence of mental health problems during the COVID-19 pandemic [53,54,55].

These findings indicated that the pandemic, caused by the spread of SARS-CoV-2 (COVID-19), as an acute public health problem, requires ongoing, comprehensive, and long-term health education to effectively mitigate panic and fear in women, thereby improving the ability of such a vulnerable population to respond in the future in a similar context.

Research on postnatal depression is challenging due to the complexity of the factors involved; therefore, some limitations of the study should also be considered. First, the questionnaire was also completed online via the Google Forms application and promoted via social media, a sampling technique that carries an inherent risk in terms of meeting the selection criteria of the study population. However, online surveys are also considered a good method of population recruitment for epidemiological research, especially in pandemic settings, and internet use is high among women of childbearing age in Europe [56,57]. Compared with national birth data, the enrolled participants were predominantly primiparous, highly educated, married, from urban areas, with a good standard of living, and without a significant personal pathological history. Given that women with higher education and the support of a partner tend to experience fewer symptoms of postpartum depression [58,59], while those with a lower level of education were susceptible to the appearance of postnatal suicide ideation [60], the high prevalence observed in our sample could shed light on the impact of the pandemic. It is also possible that mothers with marked anxiety and more severe symptoms of postnatal depressive disorder may not have completed the online questionnaire, and the likelihood of their being caught in the course of those approached during hospitalization is uncertain. Therefore, the high prevalence of postnatal depressive disorder observed in the study population may still reflect an underestimation of the pandemic situation in the general population. Second, the lack of a comparison group, as well as the cross-sectional study design, prevented us from drawing conclusions about the long-term consequences of the pandemic (whether the mental distress noted would subside in the short term or persist for a longer period of time). Such research requires longitudinal studies, conducted over several years, which are, therefore, more costly and time-consuming but which demonstrate the validity of the results. Third, recruiting patients in the postnatal period of pandemic times has been challenging because of the vulnerable terrain faced by new mothers, requiring extra involvement, patience, empathy, and extra attention from researchers. Ultimately, regression models only observed associations between socio-demographic conditions, health status, and a few obstetric features of the patients. While there may be some limitations in the study design and methods used, these inherent limitations by no means compromise the results reported. Although our data were based on a cross-sectional design, future studies may consider other methods, such as prospective designs, in-person interviews, or other objective measures, to avoid such limitations.

This study aimed to investigate the role of several maternal health status and socio-demographic factors in the risk of developing postpartum depression, yet the data were collected during the COVID-19 pandemic. We did not have information considering the impact of the pandemic on the risk of postpartum depression development among our study sample, as the design of the study was cross-sectional, and we did not have baseline data for the levels of postpartum depression risk in the pre-pandemic period. The COVID-19 pandemic may have played a major role in the risk of postpartum depression detected in this study.

We strongly encourage increased national attention to improve maternal mental health and reproductive health. Attention from primary care providers and other healthcare professionals and policymakers working in the reproductive and maternal health fields is also required [61]. Future studies should aim to examine what other variables may influence women’s psychological well-being, including the impact of the COVID-19 pandemic. Postpartum depressive disorder is known to have multifactorial causes, with the substrate being generated by a combination of factors: biological, psychological, and social. Biological factors (hormonal changes, genetic predisposition, or neurochemical imbalances) interact with psychosocial factors (stress, lack of family support, or traumatic life events); this multifactorial nature causes it to be difficult to isolate specific causes or determine the relative contributions of each factor. It may also be of value to consider the major impact of a difficult birth experience (birth trauma) and the potential emotional impact of breastfeeding success and maternal wellness in the context of perinatal mental health issues.

Despite these difficulties, research efforts are aimed at improving the understanding of postpartum depressive disorder, the accuracy and precision of the diagnosis, and the development of effective interventions to reduce the stigma associated with the condition.

## 5. Conclusions

Women who gave birth during the COVID-19 pandemic are part of a susceptible, high-risk group that should be closely monitored to minimize the effects of possible undiagnosed postnatal depressive disorder. The conducted research indicated that the pandemic caused by the spread of SARS-CoV-2 (COVID-19), as an acute public health problem, shows an alarming increase in the prevalence of postpartum depression compared with studies conducted in the pre-pandemic period. As noted, the onset of the pandemic has generated major changes in postpartum care and created new challenges that could negatively impact maternal mental health. The effects of the pandemic on mental health are of particular concern for women in the first year after childbirth. Observing these challenges and developing effective measures to prepare our health system early can be of great help for similar situations in the future. This will help and facilitate effective mental health screening for postpartum women, promoting maternal and child health. Therefore, more research is needed to understand the relationship between COVID-19 and postnatal depressive disorder.

## Figures and Tables

**Figure 1 healthcare-11-02857-f001:**
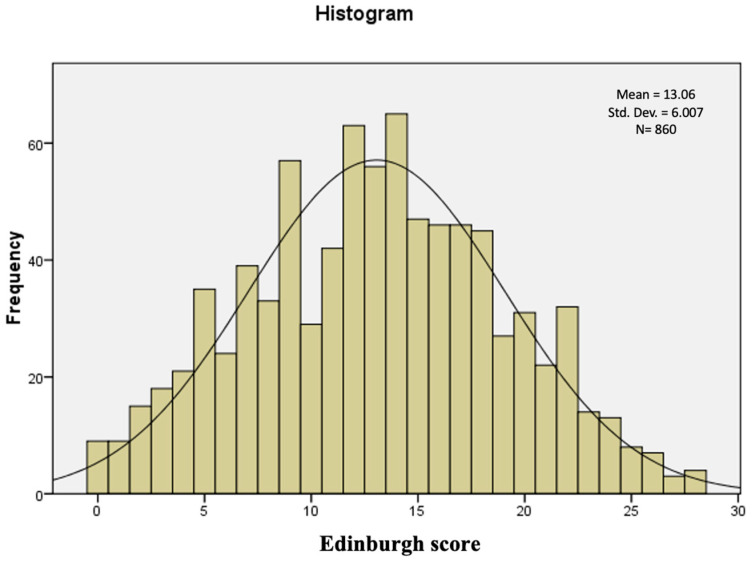
Histogram of Edinburgh scores.

**Figure 2 healthcare-11-02857-f002:**
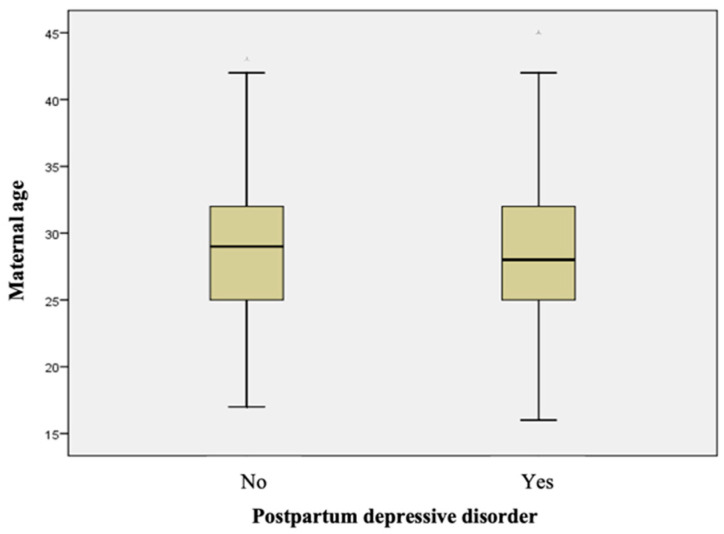
Boxplot representation for maternal age, comparative, according to depressive disorder.

**Figure 3 healthcare-11-02857-f003:**
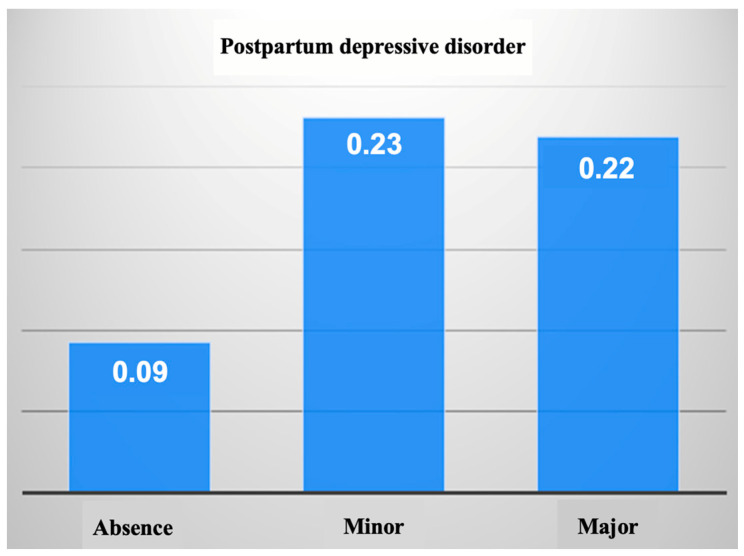
Mean values of the number of abortions on demand compared by severity of depressive disorder.

**Table 1 healthcare-11-02857-t001:** Distribution of socio-demographic characteristics and obstetric indicators among study participants.

**Age**	
Range	18–45
Mean ± SD	28.52 (4.93)
**Marital status**	
Married	86.7% (746)
Cohabiting	11.4% (98)
Single	1.9% (16)
**Area of residence**	
Urban	69.5% (598)
Rural	30.5% (262)
**Level of education**	
Higher education	59.2% (509)
High school	32.2% (277)
Primary education	8.6% (74)
**Socio-economic conditions**	
Very good standard of living	19.4% (167)
Good standard of living	59% (507)
Satisfactory conditions	18.6% (160)
Poor living conditions	3% (26)
**Workplace hazard**	
High	8.4% (72)
Medium	22.7% (195)
Low	69% (593)
**Health status**	
Good	79.4% (683)
Satisfactory	19.5% (168)
Poor	1% (9)
**Parity**	
Primiparous	65.2% (561)
Secondiparous	28.1% (242)
Tertiparous	5.2% (45)
Quarteparous	1.3% (11)
Quintiparous	0.1% (1)
**Number of miscarriages in their personal obstetric history**	
No miscarriage	80.1% (689)
One miscarriage	15.1% (130)
Two miscarriages	4% (34)
Three miscarriages	0.6% (5)
Four miscarriages	0.2% (2)
**Number of abortions performed upon request in their personal obstetric history**	
No abortion on request	87.2% (750)
One abortion on request	9% (77)
Two abortions on request	2.7% (23)
Three abortions on request	0.9% (8)
Four abortions on request	0% (0)
Five abortions on request	0.2% (2)
**Method of achieving pregnancy**	
Naturally	95.8% (824)
In vitro fertilization	1.2% (10)
With previous treatment	3% (26)

**Table 2 healthcare-11-02857-t002:** Percentage distribution of responses to the 10 questions of EPDS questionnaire.

Question Number	Item Question	Score 0	Score 1	Score 2	Score 3
Question 1	“I have been able to laugh and see the funny side of things.”	57.2% (492)	34.1% (293)	7.3% (63)	1.4% (12)
Question 2	“I have looked forward with enjoyment to things.”	65.9% (567)	22.4% (193)	9.4% (81)	2.2% (19)
Question 3	“I have blamed myself unnecessarily when things went wrong.”	7.4% (64)	24% (206)	41.4% (356)	27.2% (234)
Question 4	“I have been anxious or worried for no good reason.”	10% (86)	7.9% (68)	58.5% (503)	23.6% (203)
Question 5	“I have felt scared or panicky for no very good reason.”	15.6% (134)	21% (181)	42.8% (368)	20.6% (177)
Question 6	“Things have been getting on top of me.”	7.4% (64)	21% (181)	58.3% (501)	13.3% (114)
Question 7	“I have been so unhappy that I have had difficulty sleeping.”	31.5% (271)	24.4% (210)	32.9% (283)	11.2% (96)
Question 8	“I have felt sad or miserable.”	19.3% (166)	38.6% (332)	30.7% (264)	11.4% (98)
Question 9	“I have been so unhappy that I have been crying.”	15% (129)	16.9% (145)	46.9% (403)	21.3% (183)
Question 10	“The thought of harming myself has occurred to me.”	71.3% (613)	13.8% (119)	12.2% (105)	2.7% (23)

**Table 3 healthcare-11-02857-t003:** Comparative percentage representation of cases by socio-demographic variables, health status, type of birth, and occurrence of depressive disorder.

Association Variables		Type of Birth		*p*-Value	Postpartum Depression		*p*-Value
		Caesarean Section	Vaginal Delivery		Absence	Present	
Depressive disorder	Without	128 (25.80%)	132 (36.30%)	0.0012 *			
	Minor	78 (15.70%)	56 (15.40%)	0.98		-	
	Major	290 (58.50%)	176 (48.40%)	0.0041 *			
Marital status	Married	441 (88.90%)	305 (83.80%)	0.0382 *	229 (88.10%)	517 (86.17%)	0.511
	Cohabiting	41 (8.30%)	57 (15.70%)	0.0011 *	26 (10.00%)	72 (12.00%)	0.465
	Single	14 (2.80%)	2 (0.50%)	0.026 *	5 (1.90%)	11 (1.83%)	0.837
Education level	Less than high school	24 (4.80%)	50 (13.70%)	<0.001 *	24 (9.20%)	50 (8.30%)	0.764
	High school graduate	160 (32.30%)	117 (32.10%)	0.991	67 (25.8%)	210 (35.00%)	0.010 *
	Higher education	312 (62.90%)	197 (54.10%)	0.012 *	169 (65.00%)	340 (56.70%)	0.028 *
Socio-economic conditions	Good	290 (58.50%)	217 (59.60%)	0.799	157 (60.40%)	350 (58.30%)	0.617
	Very good	111 (22.40%)	56 (15.40%)	0.013 *	60 (23.10%)	107 (17.80%)	0.087
	Poor	10 (2.00%)	16 (4.40%)	0.067	4 (1.50%)	22 (3.70%)	0.131
	Satisfactory	85 (17.10%)	75 (20.60%)	0.224	39 (15.00%)	121 (20.20%)	0.088
Health status	Good	386 (77.80%)	297 (81.60%)	0.202	232 (89.20%)	451 (75.20%)	<0.001 *
	Poor	6 (1.20%)	3 (0.80%)	0.816	1 (0.40%)	8 (1.30%)	0.404
	Fair	104 (21.00%)	64 (17.60%)	0.248	27 (10.40%)	141 (23.50%)	<0.001 *

*—Significant difference.

**Table 4 healthcare-11-02857-t004:** Descriptive statistics of numerical variables, comparatively, by depressive disorder.

Variable	Depressive Disorder	N	Mean	Std. Deviation	Std. Error Mean	Mean Rank
Maternal age	No	260	29.04	4.782	0.297	459.31
	Yes	600	28.29	4.982	0.203	418.01
Number of miscarriages	No	260	0.20	0.511	0.032	413.76
	Yes	600	0.28	0.602	0.025	437.75
Number of abortions on demand	No	260	0.09	0.328	0.020	409.48
	Yes	600	0.22	0.622	0.025	439.61
Number of births	No	260	1.43	0.639	0.040	437.21
	Yes	600	1.43	0.675	0.028	427.59
Number of miscarriages	Absence	260	0.20	0.511	0.032	413.76
	Minor	134	0.34	0.660	0.057	454.24
	Major	466	0.27	0.585	0.027	433.01
Number of abortions on demand	Absence	260	0.09	0.328	0.020	409.48
Number of births	Minor	134	0.23	0.612	0.053	443.26
	Major	466	0.22	0.625	0.029	438.56
	Absence	260	1.43	0.639	0.040	437.21
	Minor	134	1.54	0.752	0.065	462.49
	Major	466	1.39	0.648	0.030	417.56

**Table 5 healthcare-11-02857-t005:** Descriptive statistics of the Edinburgh score, comparative, by type of birth.

Variable	Type of Birth	N	Mean	Std. Deviation	Std. Error Mean	Mean Rank
Edinburgh Score	Vaginal delivery	364	12.46	6.116	0.321	449.71
	Caesarean section	496	13.51	5.892	0.265	404.32

## Data Availability

The data presented in this study are available on request from the corresponding author.

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
