# Peer review of "The Impact of the COVID-19 Pandemic on Depressive Disorder with Postpartum Onset: A Cross-Sectional Study"

_healthcare, 2023, doi:10.3390/healthcare11212857_

Round 1

Reviewer 1 Report

Comments and Suggestions for Authors

Healthcare Journal

Manuscript: The Impact of the COVID-19 Pandemic on Depressive Disorder 2 with Postpartum Onset

The manuscript describes the impact of the COVID-19 pandemic on postpartum depression in a sample of 860 women. The assessment was performed in the Obstetrics and Gynaecology Clinical Departments I and II and online using Google Forms. The scale used to assess depression was the Edinburgh Postnatal Depression Rating Scale (EPDS). Results showed the relationships between sociodemographic, economic, educative, birth conditions, and depression. An important increase in the percentages of postpartum depression was found.

I find the research very relevant in the light of the impact of COVID-19 pandemic on women, mainly in those who must go to hospital to give birth when the possibility of COVID-19 infection was very high in a hospital setting.

The authors describe the EPDS questionnaire presenting the three factors that this questionnaire assess: depression, anxiety, and suicide, but there is no information about these factors in the Results and Discussion sections. All the results presented and discussed are those of the total score of the EPDS. Considering that just a single questionnaire is used in the study, it is a pity that the information about these three separate factors is not presented nor discussed. My recommendation is that the authors include the most relevant information obtained from these separated factors in the Results and Discussion sections of the manuscript.

The fact that the partum (birth) took place in hospitals in the middle of the COVID-19 pandemic, were COVID-19 patients were probably being treated, with the corresponding increased risk of infection for the future mothers, is not presented strongly enough in the manuscript. I think that, depending on the results of the anxiety factor, this issue could be included in the Discussion section.

Also, the following changes may improve, in my opinion, the manuscript:

INTRODUCTION

1.  Several sentences in the introduction need citations:  Lines 66-71; Lines 72-75; Lines 88-94.

2. Authors state in line 109: “The aim of this study was to assess the mental health status ….” but only depressive symptoms were assessed. I think it is more accurate to write “the aim of the study was to assess depressive symptoms…….”

MATERIALS AND METHODS

1: Change the “Studied Population” subsection by “Sample description” or “Participants”. The study is not assessing a “population” but a “sample”.

2. Lines 135-137: “The questionnaire was completed both in the Obstetrics and Gynaecology Clinical Departments I and II and online using Google Forms”, percentages of participants that completed the questionnaire online and in situ should be included.

3. Lines 154-157: Is it there any study that has validated the EPDS questionnaire in Romania, or studies that used the EPDS in Romanian samples? If yes, please include the information. If not, please explain if there was a translation or any other method to adapt the questionnaire to the sample.

4. Lines 165-169. Citations are needed. “Validity studies show that the 168 scale can correctly identify 92.3% of women with postpartum depression” Which studies? Please cite them.

5. A new paragraph should be included in the Materials and Methods section (separated from the studied population subsection) giving information about the socio-demographic, economical, educative, and birth conditions variables.

RESULTS

1. Line 174, 223, 228, etc. Please use the same way to name χ2.

DISCUSSION

A more elaborate discussion is need, and not just a repeat of the description of the results found.

Reviewer 2 Report

Comments and Suggestions for Authors

Dear authors,

Thank you for your contribution to a deeper understanding of effects from the pandemic. I generally find your study of interest. However, I have some major concerns that needs to be addressed.

1.       My first question is if you had ethical permission from ethical board for this study, since it consists from sensitive data and a vulnerable group of study participants? This is not stated in the paper and needs to be addressed.

2.       In the background, there are sections that are repeated. For example, in line 113 and forward, you repeat from the introduction. Please make sure that the background in concise and clear.

3.       The tempus (language) is different during the whole paper. Please make sure that the grammar is correct, incl tempus.

4.       Did the study include all mothers (except from those under the age of 18) or a sample?

5.       To ease the reading of your demographics and descriptive results, consider to present these as a table instead of a long text.

6.       Consider to present the results now presented in Figure 2,3,4,5 as a Tables instead, or in the same table, and add p values for statistical differences for each row. In Table 1, you say that “the association is significant” but you don’t say which of the associations that is significant? The table presents several possible associations.

7.       Your main conclusion is that the pandemic had impact on the wellbeing of mothers. However, you have not put your results or data in relation to before or after the pandemic, why I think this is not a correct conclusion.
